# Toward a Population-Based Breast Cancer Risk Stratification Approach? The Needs and Concerns of Healthcare Providers

**DOI:** 10.3390/jpm11060540

**Published:** 2021-06-10

**Authors:** Jolyane Blouin-Bougie, Nabil Amara, Jacques Simard

**Affiliations:** 1Department of Management, Faculty of Business Administration, Laval University, Québec City, QC G1A 0A6, Canada; nabil.amara@fsa.ulaval.ca; 2CHU de Québec-Université Laval Research Center, Québec City, QC G1V 2G4, Canada; jacques.simard@crchudequebec.ulaval.ca; 3Department of Molecular Medicine, Faculty of Medicine, Laval University, Québec City, QC G1A 0A6, Canada

**Keywords:** breast cancer, risk stratification, healthcare providers, perceptions, interviews, knowledge value chain, genetic counselling

## Abstract

Given the expanding knowledge base in cancer genomics, risk-based screening is among the promising avenues to improve breast cancer (BC) prevention and early detection at the population level. Semi-structured interviews were conducted to explore the perceptions of healthcare professionals (HPs) regarding the implementation of such an approach and identify tools that can support HPs. After undertaking an in-depth thematic content analysis of the responses, 11 themes were identified. These were embedded into a logical model to distinguish the potential eligible participants (who?), the main clinical activities (how?) and associated tools (what?), the key factors of acceptability (which?), and the expected effects of the strategy (why?). Overall, it was found that the respondents positively welcomed the implementation of this strategy and agreed on some of the benefits that could accrue to women from tailored risk-based screening. Some important elements, however, deserve clarification. The results also highlight three main conditions that should be met to foster the acceptability of BC risk stratification: respecting the principle of equity, paying special attention to knowledge management, and rethinking human resources to capitalize on the strengths of the current workforce. Because the functioning of BC risk-based screening is not yet well defined, important planning work is required before advancing this organizational innovation, and outstanding issues must be resolved to get HPs on board.

## 1. Introduction

The recent evolution of genomic technologies and research has led to the discovery of diverse genetic variants associated with BC risk. These may confer high or moderate risks of developing BC. Alternatively, some polymorphisms (SNPs) may confer a low although clinically relevant risk when combined into a polygenic risk score [1,2]. The idea of adapting the current BC screening programs on the basis of different personalized risk levels emerged from these discoveries. Called BC risk stratification, such a service is viewed as a step toward the ultimate objective of personalized medicine by incorporating genomic innovations and tailoring preventive and early detection interventions to different risk groups. Implementing BC risk stratification at a population-based level is, however, a challenge. This “approach has not yet been implemented anywhere in the world” [3], and there are only rare tentative services that can be studied to concretely inform stakeholders about the advantages and disadvantages of this strategy in real settings. To inform researchers involved in the development of risk-based screening, and managers and decision makers interested in innovations in BC prevention, this study aimed to explore the perceptions of healthcare professionals (HPs) regarding BC risk stratification implementation.

BC risk stratification is an organizational innovation [4] intending to adapt the content of the genetic counselling process to the organizational framework of BC screening. It is a new means of tackling BC prevention and early detection [5] that lies between the well-known BC screening program and genetic counselling services [6,7]. It is expected that women’s personal BC risk will first be assessed. This will include a collection of medical, personal and familial history, a genetic test, and an estimation of BC risk level, by combining genetic and non-genetic risk factors with the support of a risk prediction model (RPM). Second, the BC risk will be communicated to women conjointly with a discussion about the benefits and harms of choosing one of the recommended risk management strategies, according to their risk level.

Overall, the literature on BC risk stratification is optimistic. Many clinical benefits are expected [5,6,8]: tailoring risk management strategies to each woman or risk group, adapting BC prevention to women’s needs, detecting more BCs at an earlier stage, identifying more high-risk women, and improving the usefulness of prevention modalities and precision of patients’ care and follow-up. Moreover, recent studies, in which the cost effectiveness of tailoring screening by BC risk level is assessed using hypothetical cohorts or simulation modelling, suggest risk stratification can improve the benefit-to-harm ratio and reduce costs of BC screening [9,10,11]. Others have also suggested the feasibility of some components of this approach in real settings. For instance, it is feasible to incorporate the use of risk prediction forms and then to adapt screening intervals [12]; to use decision aids to induce more realistic expectations and informed decisions regarding BC screening modalities [13]; or to use postal invitations to sensitize women to BC risk assessment in order to identify more at-risk women [14]. Additionally, positive developments have recently been made, such as polygenic risk scores that combine the multiple effects of SNPs [2,15,16] and RPMs for BC population-based risk assessment [6,17]. Polygenic risk scores and RMPs are at the heart of the strategy and concern, respectively, the challenge of determining the variants to consider in a genetic test, and the combination of modifiable and non-modifiable risk factors to be used in RPMs, for a precise and high-quality assessment of BC risk [18,19,20].

Nevertheless, enthusiasm about BC risk stratification also faces a number of structural challenges. First, although few general organizational features are expected and the adaptation of BC screening programs is the most cited solution, no precise business model is available for replication. The overall functioning of this approach, healthcare trajectories, and pathways (e.g., participation protocol, procedure for evaluation or risk re-evaluation) must be established for the feasibility to be precisely determined [6,7,20,21]. Second, it implies numerous novelties (e.g., genetic tests, polygenic risk scores, RPMs, guidelines, responsibilities) that will be required to be simultaneously adopted by a large range of stakeholders, which make it more complex to implement. Despite a large informative body of literature on genetic counselling for BC in primary care and on the prevention of BC in high-risk women, few studies have directly elicited the views of HPs regarding the potential implementation of an approach using BC risk stratification [22,23]. Third, it is known that organizations and HPs will need support, training, and tools to manage the risk stratification process [7,21,22]. However, these should be determined by taking into account the context in which the approach will be implemented [6,20]. Finally, HPs must perceive the benefits of the approach and consider it to be acceptable. Nevertheless, evidence on this matter is lacking [6,7,8,24].

Because there is increasing international interest in risk-based screening for BC [25], it is an opportune time to share the results of this study, which aimed to shed light on the perceptions of HPs regarding the implementation of a BC risk stratification population-based approach in Québec (Canada); and to identify actions, strategies, and tools that can be developed to support HPs in risk-based screening for BC. A qualitative explorative study design was developed to meet these objectives.

## 2. Materials and Methods

The population under study comprised HPs directly involved in BC genetic counselling or screening services in the province of Québec (Canada). To obtain a purposive sample of respondents, the snowball tactic was used to identify potential respondents, which were included on the basis of inclusion criteria (Table 1). To ensure the diversity of information provided by the respondents, the region, practice settings, and specialties of the HPs were considered. The number of recruited respondents was predetermined to be around twenty. This choice was made according to a known indicator based on a prior study of Guest, Bunce and Johnson [26]. The authors demonstrated that data saturation was generally reached with 6 to 12 respondents, when the sample is relatively homogeneous, the structure of the interview is respected for all participants, and the interviewees are specialized in their domain. This was the case in the current study.

A semi-structured interview guide was developed, validated by an interdisciplinary expert committee, before being pre-tested with a practicing oncologist. This contained three categories of questions addressing: general perceptions regarding the implementation of BC risk stratification; perceived level of knowledge on BC genetics; and current clinical practices, needs in terms of tools or resources, and anticipated changes. To inform this latter section of the guide, the four main activities of *The breast cancer genetic counselling process framework* [27] (risk assessment, genetic testing investigation, risk communication, and risk management) were used to develop questions. This model was chosen because it represents the general process of genetic counselling and the expected general content of the BC risk stratification approach.

Data were collected between October 2014 and February 2015. Overall, 65 HP were contacted, 18 agreed to participate, and 15 valid interviews were transcribed for analysis. Two audio files failed to be recorded properly, and one participant was finally not interviewed as planned. All interviews were conducted by phone. At the beginning of the interview, participants were briefly informed about BC risk stratification. Data were anonymized before the transcription of verbatim responses and were revised, in addition to the audio files, prior to the analysis to ensure their reliability. All material was initially gathered in French and freely translated into English for publication purposes.

The content thematic analysis followed a three-step process [28]. First, a deductive and descriptive codification was undertaken according to the questions of the interview guide. Large blocks of text were coded under each underlying question’s topics to create a first set of nodes. Then, each block of text was decomposed to identify all ideas proposed by respondents, and to create representative sub-nodes with relatively small quotes or even sentences in taking into account the context of the excerpts. Second, to identify patterns, the nodes and quotes were revised and grouped following the techniques of repetition and chunking and sorting. Via these sub-coding steps, the list of nodes was refined and aggregated into themes representing larger categories that comprised a set of sub-nodes. These two steps were undertaken with the support of an assistant researcher (KL). Uncertainties were discussed and the hierarchical tree of nodes was entirely revised to ensure quality and validity. Third, to highlight links between themes and improve the usability of the results, themes were categorized into five dimensions (meta-themes) representing a component of a logical model: “who” (targeted population); “how” (particular clinical activities); “what” (associated tools and tactics); “which” (prerequisites or conditions); and “why” (anticipated effects). The analysis was stopped when all quotes were categorized into a well-defined node. The analysis was performed using NVivo 11 [29] and led by the same researcher who handled the interviews (JBB).

## 3. Results

Overall, 15 interviews with an average length of 52 min were analyzed. Table 2 provides a summary of respondents’ characteristics. Interviewees were mostly women working in university hospitals of the metropolitan region. They were relatively equally distributed regarding their other characteristics.

Eleven themes were identified; these are reported in Table 3, in addition to their degree of usability. The hierarchical tree of nodes (Appendix A) and multiple vivid quotes (Appendix A) are available as Appendix A.

### 3.1. WHO?: Eligible Participants

Most respondents (10 out of 15) acknowledge some women might benefit from BC prevention interventions outside the age range of 50-69 (eligibility criterion for the *Programme québécois de dépistage du cancer du sein*, the current breast cancer screening program in Québec). Nonetheless, the huge number of women to screen is perceived to be an impossible challenge to overcome. Nine out of ten respondents made several comparisons with the current screening program, to express the difficulties in replicating this model. No preponderant opinions about potential eligible participants emerged, but some (4 out of 10) argued it should be accessible to all women for ethical concerns.

### 3.2. HOW and WHAT?: General Clinical Activities and Associated Tools

Four general activities were distinguished: identification and invitations, risk assessment, risk communication, and risk management and follow-up. The first emerged from the data, whereas the latter three were among the main steps of the theoretical model used to build the interview guide.

#### 3.2.1. Identification and Invitation

The approach needed to invite potential eligible women remained unclear to respondents (14 out of 15); however, they proposed solutions based on available recognized BC risk factors that can be easily collected to pre-select women to be invited. The most recurrent of these were family history, breast density, and age. Other suggestions to complete this activity were to use a self-administered questionnaire, to use networks and family members as information relays, or to invite women via letter like to the current screening program.

#### 3.2.2. Risk Assessment

All respondents (15 out of 15) talked about the usefulness of a clinical questionnaire to assess BC risk and determine the need for further evaluations or genetic testing. This appears to be well integrated into their practice, but the collection of risk factors, notably family history, varies across settings, specialties, and HPs. The majority (14 out of 15) of respondents pronounced themselves in favor of a standardized and computerized tool for BC risk assessment. However, general practitioners were worried about the complexity and time required to use such tools. In contrast, genetic counsellors indicated they find RMPs generally easy to use, but not necessarily essential, because they often considered their experience and clinical judgement to be sufficient. Rather, they were concerned about the relevance of available RPMs and which of these to use for a particular patient.

#### 3.2.3. Risk Communication

Respondents mostly noted the lack of RCTs (14 out of 15). Indeed, HPs indicated they provided verbal explanations to patients (13 out of 15), without necessarily using graphs or visual representations of risk, nor providing written information to patients. They wanted more RCTs, written or visual, that could be adapted to each patient, improve patients’ retention of information, and increase awareness regarding BC prevention. A centralized computerized information resources center, in which recognized, standardized, and simplified information on BC risk is provided, was a practical solution proposed by a respondent that could be useful for HPs and patients.

#### 3.2.4. Risk Management

HPs generally combined diverse elements before making recommendations to their patients; in particular, these elements include patients’ risk level, available clinical guidelines, and patients’ desires and preferences. However, many respondents (9 out of 14) deplored the lack of standardized clinical guidelines, notably for young women. They often called on their colleagues’ expertise, professional experience, and clinical judgement to make final risk management recommendations to their patients. Respondents indicated that the ideal approach would be the inclusion of uniform criteria and recommendations in the results provided by a RPM or a set of tools adapted for BC risk stratification to the context in Québec.

### 3.3. WHICH?: Prerequisites or Conditions

#### 3.3.1. Ethical Principles

Respondents mostly discussed the equity of access to care (12 out of 14). This was not unconnected to the current lack of resources in the healthcare system. Indeed, respondents expressed concerns about how resources would be equally and efficiently used. As underlined by a respondent, some women recognized to be at-risk (e.g., mutation carriers on PALB2) are not provided with access to services and some ill women have difficulties accessing radiological modalities. The number of available hours for BC screening, by MRI, ultrasounds, or mammograms, in hospitals is currently insufficient to meet demand, and respondents believe this problem could be exacerbated if BC risk stratification is implemented. Moreover, dedicating time to BC prevention, when some organizations can barely meet the needs of urgent cases, was perceived negatively by respondents. Some interviewees explicitly highlighted that patients’ access to radiological modalities may vary according to the settings in which their prescribing physicians work; this was another source of inequity.

#### 3.3.2. Services Organization

Given the functioning of a BC risk stratification approach is not yet well defined, respondents expressed concerns about its feasibility (9 out of 13). They indicated the need to involve HPs during the developmental and implementation phases of the project, to reflect their reality as much as possible (e.g., pre-testing tools, running pilot projects). Seven out of thirteen respondents also emphasized the fact BC risk stratification must be complementary to the current screening program and well integrated into existing services. No confusion or overlaps must exist between these two approaches for HPs to be comfortable with the services’ trajectory.

#### 3.3.3. Human Resources Administration

Almost all interviewees raised the issue of the scarcity of human resources (12 out of 14), notably in genetics (8 out of 12). Many suggested more collaboration with HPs specialized in genetics and called for more resources, particularly in rural regions, for the approach to be ethically acceptable and feasible. In addition, a matter of contention among respondents related to the role and responsibility of HPs regarding at-risk patients. Genetic counsellors embraced their role with high-risk women, but not with those of the general population. In contrast, general practitioners thought that assessing and managing risk is “more the domain of medical genetics”. Despite 13 out of 14 respondents recognized that BC risk stratification activities should be conducted by general practitioners, they acknowledged time was lacking to fully endorse this new role.

#### 3.3.4. Knowledge Management

The general practitioners’ lack of knowledge in genetics was highlighted, but some specialists admitted that onco-genetics have become more complex, making it more difficult to keep up to date. As one geneticist said, general genetic knowledge should be developed and made more accessible to all physicians. Moreover, 11 out of 15 respondents asked for a variety of knowledge exchange tools (e.g., facilitating references from one setting to another, having electronic medical records), going beyond the clinical activities discussed above. Some (5 out of 15) also underlined the fuzziness of the available evidence-based knowledge on BC screening and the multiplicity of guidelines, which contribute to variations in practices among regions and settings. Finally, they emphasized (9 out of 15) the importance of the diffusion activities, notably to sensitize HP, and to increase public awareness about BC risk and the new modalities to be proposed.

### 3.4. WHY?: Anticipated Effects

#### 3.4.1. Patients

Respondents mostly noted the negative psychological impacts of the implementation of BC risk stratification (9 out of 11) on patients (e.g., anxiety, apprehension, or worries) that could lead to unnecessary complementary clinical examinations. In contrast, patients’ awareness and empowerment were seen to be potential positive effects by a number of respondents (5 out of 11).

#### 3.4.2. Services

Thirteen out of fifteen respondents expected that BC risk stratification would improve the quality of services in BC prevention (e.g., improve the accuracy of risk assessment, facilitate identification of at-risk women, personalize patient management and follow-up). Half of the respondents (7 out of 15) also saw the standardization of the services as an important advantage. Many respondents (8 out of 15), however, were concerned about the increasing demand for screening tests and the pressure this would place on health organizations (e.g., hospitals, breast clinics). The potential for rising costs and an extended waiting list for screening modalities were the negative impacts cited by respondents.

### 3.5. Results Schematization

The main results discussed above are summarized in Figure 1. The figure emphasizes the issues, concerns, needs, and benefits from the perspective of respondents. These elements were embedded in a logical model to highlight the links between each of its components (who, how, what, which and why).

## 4. Discussion

This study sheds light on the perceptions of HPs regarding the implementation of a BC risk stratification approach in the province of Québec (Canada). Overall, the respondents welcomed the implementation of this strategy and agreed about some of the benefits that could result for women from tailored risk-based screening. Some important elements, however, deserve clarification (e.g., eligible participants, roles and responsibilities of clinicians, direct positive impacts on patients). Other studies using this approach in Québec’s context have reported similar results [23,30]. This study also allows identification of tools considered by HPs to be potentially useful for managing BC risk stratification. Finally, the present work enables identification of conditions to foster the acceptability of BC risk stratification, notably, respecting the principle of equity for all activities of the value chain, paying special attention to knowledge management that extends beyond the clinical level to include strategic and marketing components, and rethinking the roles and responsibilities of human resources to capitalize on the strengths of the current workforce.

### 4.1. Ensuring Ethical and Standardized Services

For all of the clinical activities of the BC risk stratification approach, from identification to risk management, respondents were concerned about the principle of equity. They questioned the feasibility of including all women in such a program, given the resources available, similar to the managers of the *Programme québecois de dépistage du cancer du sein* interviewed by Hagan et al. [30]. One of their solutions was to add personal risk factors as eligibility criteria to lower the number of women to be screened. The inability of all women to benefit from these personalized services can be seen as a problem and an obstacle to the program’s acceptability [20]. Given population screening programs are, in essence, inclusive models of prevention and surveillance, the possible move toward a restrictive risk-based screening program [19] is likely to conflict with the basic principles of accessibility and equity of the BC screening program in Québec [31]. Adding eligibility criteria that are unknown to healthcare authorities also complicates women’s identification through registries and invitation by letter. Nonetheless, this solution could help to focus on women for whom screening works better, or provides more benefits (e.g., detection of malignant lesions early) and fewer downsides (e.g., false positives) [32]. This would then imply not seeing equity in terms of access to care for all, but instead as “equal management for equal risk” [33]. Furthermore, it is known that not everyone benefits equally from screening [32]. Thus, experts and authorities must agree on the targeted population that will have access to risk-based screening [6], which is tied to the information used to assess BC risk. It is important to note that the method of identification and invitation can have an impact on fair access to and participation in screening [3,24].

Another legitimate concern relates to the use of RPMs. A variety of RPMs exist that “…use different input parameters and different outcomes and were developed and validated in different populations” [19]. Most of these also need to be adapted for use at the population level [6,20,34]. These considerations, in conjunction with the demands of the respondents for standardization of practices and equity, would imply choosing one RPM that is sufficiently flexible to evaluate the BC risk of all eligible women in the province, regardless of their age group, ethnicity or a priori risk level. The model should also be sufficiently accurate to allow different risk groups for women to be individually and correctly oriented to relevant preventive interventions, according to their respective calculated risk. Currently, the latest version of BOADICEA [17] is being tested by the researchers of PERSPECTIVE I&I (https://etudeperspective.ca/ [accessed on 5 May 2021]) in Québec and Ontario (Canada) [35].

A third ethical concern of respondents relates to the disparity of the recommendations among providers and health organizations. Practice variations are also known in the integration of genetic services in primary care [36], the management of BRCA mutation carriers [37], and for women referred for mammography [38]. One explanation could relate to a variation of practices in the upstream process [39]. In addition, because the use of patient-centered care is increasing in healthcare [5], it could also be due to women’s choices. The respondents mentioned related issues that could partly explain risk management variations and inequities: difficulties of accessing radiological modalities in a timely manner; multiplicity of guidelines on BC screening; and lack of clear recommendations for some subgroups, such as young women. HPs evidently want clear, accessible, and standardized recommendations regarding early detection modalities and risk management interventions by risk level and patient subgroup.

### 4.2. Focusing on Knowledge Management

Numerous tools and tactics were proposed by the respondents, and several of them aimed to support HPs in their expected new role. Thus, they reflect their needs in terms of learning and assistance to provide personalized services of high quality, which was highlighted in previous studies [21,22,38]. As a result, consideration can be given to integrating tools and tactics into a sound knowledge management plan that is aligned with the components of a BC risk stratification approach. For changing work behaviors, processes, and even culture in organizations, knowledge management experts “say that technology is 10% of the effort required; process is 20% and 70% being people/cultural issues” [40]. It is thus crucial that particular attention be given to the intended users (i.e., HPs). A number of means can be applied for this purpose. As mentioned by some respondents, to involve the intended users in the development of tools can increase the fit between their needs and their professional reality [6]. This approach also provides a ways to connect people to knowledge, and users to developers. Another means could be to assess the gap between the knowledge and skills of HPs, and those needed to develop targeted educational material or knowledge-sharing mechanisms (e.g., training, coaching, networking). The limited genetic proficiency of clinicians and the growth of the genomic knowledge base are both recognized as barriers to the adoption of genetic-related innovations [41]; this is in line with the opinions of the respondents.

Otherwise, managing knowledge implies assessing if the tools or tactics proposed add value to stakeholders [40]. They could complement each other, used as substitutes, or used independently to support a strategy. For instance, it will be essential to inform the women and the HPs about BC risk stratification [22,23]. At this time, the message of the diffusion strategy should be adapted to different public groups to foster participation [30]. Then, multiple sensitization or educational messages could co-exist, all adding value to the risk-based approach, in a substitutive manner, for various targeted groups (e.g., rural communities, young women with no prior experience of screening). Moreover, providing training appears to be essential to improve HPs’ knowledge about BC genetic risk [7,22]. This will contribute on its own (i.e., independently) to the success of the approach. Similarly, developing a RPM can support most of the BC risk stratification clinical activities, whereas leaflets can remain relevant [20,42] in fulfilling needs in terms of risk communication. The ability to determine the contributing inputs of tools proposed for a particular action, a specific step of the process, or even the overall strategy, can help to propose a parsimonious tool kit and to balance the amount of effort aimed at developing or updating tools in regards of the needs of end users.

### 4.3. Rethinking Human Resources

In addition to the lack of human resources mentioned by respondents, there is a consensus that the clinical activities of the BC risk stratification approach should be including in general practitioners’ work diaries [20]. This seems logical because general practitioners are at the forefront of the healthcare system, and are thus considered to be key to disease prevention [23,42]. This nonetheless has some managerial and clinical implications.

From a governance perspective, if general practitioners are responsible for determining participants’ eligibility and the subsequent clinical activities, the success of this approach, including duties and responsibilities, will be largely placed on the shoulders of HPs. This will challenge the Canadian management model of population-based programs, in which the responsibility usually lies with a governmental agency [43]. From a clinical perspective, HPs feel that do not have sufficient resources (e.g., time, knowledge) to fully endorse the new responsibilities associated with BC risk stratification. Similar obstacles were highlighted in other studies conducted in Québec [23,30]. Moreover, general practitioners are currently mostly involved in risk management and patients’ follow-up of the BC genetic counselling process [27]. Previous steps (i.e., risk assessment or communication) are usually managed by specialists or genetic counsellors. It is therefore not surprising that general practitioners feel uncomfortable about leading all of the clinical activities of a BC risk stratification approach.

It could be then more efficient to centralize all BC risk stratification activities within existing BC clinics rather than offering a completely different pathway [6,20,23,24]. This would allow the approach to capitalize on the current infrastructure, including the competences of their workforce; to facilitate the harmonization and respect of the core values of the current screening program; and to let the overall management and accountability of the strategy lie with the government. In addition, BC clinics are preferred over private clinics for developing interprofessional collaboration given the physical proximity of diverse types of HPs involved in BC prevention and care, and their strategic organizational structure based on the problems of BC. BC clinics can then offer all of the services of the clinical activity chain, and more easily control continuity between services, than general practitioners in private clinics.

The clinical activities chain proposed in this study only presents the general activities of a potential risk stratification approach, and not all of the tasks associated with each of these. Analyzing HPs’ roles and tasks to endorse and assess the notion of shared responsibilities of all stakeholders involved in this new approach could provide relevant indicators (e.g., needs in terms of skills’ development or services’ continuity) and help to find the right balance of roles and power-sharing between actors to favor the success of the strategy.

### 4.4. Strengths and Limitations

The results of this study should be interpreted in light of some strengths and limitations. First, a trade-off in the heterogeneity of the sample of respondents was favored to obtain the view of all types of providers involved in the BC prevention process. This led to the inclusion of either generalists, specialists, or genetic counsellors, and provided a larger range of opinions and allowed comprehensive understanding of the phenomenon to be developed. Nonetheless, there were discrepancies between the type of HP for some themes. Therefore, it would be of interest to study the perceptions of these types of HP separately or using a larger sample size to perform comparisons between them. Furthermore, the results from this study might benefit from additional research from other stakeholders’ perspectives, such as decision makers, to obtain a more in-depth political and administrative view of the implementation of this approach. Second, although this study did not uncover all possible opinions about the implementation of BC risk stratification in Québec, given the sample size, the results were grounded in the reality of HPs and may be used to determine future research directions. Third, although the data were collected six years ago, most of the subsequent changes were mentioned in the literature review. The service offering and the context of BC prevention and early detection in Québec remained the same, and there is still no service of this kind implemented in Canada. HPs may be more aware of the emerging possibilities in personalized services today than at the time of data collection, as this is increasingly discussed by researchers and health organizations. For instance, there is a new informative section for physicians on ongoing studies related to tailored risk-based screening on the Web site of the *Programme québécois de dépistage du cancer du sein* (http://www.depistagesein.ca/ [accessed on 4 June 2021]). However, HPs are a group of stakeholders for which building awareness and knowledge in personalized medicine will be complex and result from long-term efforts [21]. Thus, although this publishing delay may be a source of possible bias, the results of this study remain relevant and can help to propose context-specific solutions about the implementation of BC risk stratification. Fourth, the use of a recognized theoretical model provided a solid foundation for collecting the data from the current practices of HPs in BC prevention and onco-genetics. Similarly, the use of a general operational framework used to build programs in healthcare, a logical model, allows links between themes to be distinguished. It also increases the comprehensiveness and the usability of the study’s results. Finally, readers can judge the thoroughness of the data analysis by the use of multiple rounds and scrutiny techniques used to identify themes [44]. The presentation of multiple and vivid quotes illustrating respondents’ perspectives contributes to the credibility of the findings.

## 5. Conclusions

This study contributes to the pool of knowledge on personalized medicine and the translational issues surrounding the use of genomic information in BC prevention from the perspective of important stakeholders in the healthcare system, that is, HPs. The study highlights HPs’ perceptions regarding both current practices in BC prevention and anticipated changes about the implementation of BC risk stratification. It also reveals some of HPs’ needs and concerns. Overall, this study provides important cues for the implementation of BC risk stratification, in addition to future research directions to explore. No BC risk stratification approach exists in Canada and almost all large program parameters that concern its management, configuration, or regulation remain undefined. An important upstream task of programming and planning must be completed before advancing this idea of a new provincial BC prevention approach.

## Figures and Tables

**Figure 1 jpm-11-00540-f001:**
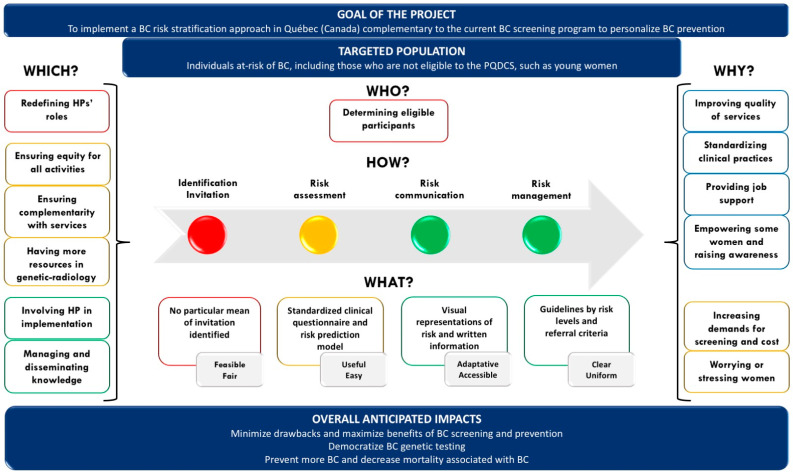
Results embedded in a logical model *. * Only information in white figures represents respondents’ answers. The contextualization elements of the study are in shown in dark blue. 
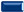
 PQDCS: Programme québécois de dépistage du cancer du sein; BC: Breast cancer; HP: Healthcare professionals; 
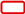
 = Issues; 
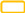
 = Concerns; 
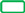
 = Needs; 
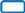
 = Benefits; 
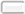
 = Clinical activities chain and main expected tools’ characteristics by activity.

**Table 1 jpm-11-00540-t001:** Inclusion criteria of potential respondents.

To be a practicing physician in the province of Québec (Canada)
2. To be a healthcare provider involved in BC risk prediction or communication or BC screening
3. To have one of the following professional titles: Geneticist, Oncologist, Radiologist, Surgeon, Genetic counsellor, General practitioner (family physician), Obstetrician–Gynecologist
4. To understand and speak French well

**Table 2 jpm-11-00540-t002:** Characteristics of the sample (*n* = 15).

*Gender*	
Men	1
Women	14
*Types of healthcare professionals*	
MD general practitioners	6
MD specialists	5
Genetic counsellors	4
*Workplace (practice settings)*	
University hospital	9
Affiliated health center	2
Regional health center	2
Family practice clinic	2
*Administrative regions*	
Capitale-Nationale	2
Montréal (metropolitan area)	7
Chaudière-Appalaches	1
Estrie	3
Saguenay/Lac-St-Jean	2
Average duration of interviews: 52 min.	

**Table 3 jpm-11-00540-t003:** Main themes, by logical model components, and their degree of usability.

Meta-Themes	Themes	Nb of Sources	Directions and Precision on Usability of Finding	Ex. of Quotes *
WHO? *targeted population*	Eligible participants	10	Ambiguous: no definite answer; concerns for feasibility and equity	Q1–Q2
HOW? *clinical activities* WHAT? *associated tools or strategies*	Identification and invitation	14	Ambiguous: no definite answer; concerns for feasibility; options to be evaluated	Q3–Q4
	Risk assessment	15	Almost clear: agreement on the tools needed (standardized clinical questionnaire and RPM); concerns about usefulness and complexity of the tools	Q5–Q6
	Risk communication	15	Clear: need more risk communication tools	Q7–Q8
	Risk management	15	Clear: agreement on the need for clear and standardized provincial criteria and recommendations	Q9–Q10
WHICH? *conditions or* *prerequisites*	Ethical approach	14	Clear: concerns about current sources of inequity—notably linked to lack of resources; expectations toward respecting the principle of equity in all activities of the approach	Q11–Q13
	Services organization	13	Ambiguous: concerns about feasibility; desire to be involved in the development and implementation	Q14
	Knowledge management	15	Clear: many tools needed that concerned learning strategies, knowledge transfer and diffusion strategies	Q15–Q17
	Human resources administration	14	Clear: concerns about lack of resources; discrepancies in regard to HPs’ roles to be resolved	Q18–Q20
WHY? *potential effects*	Patients or population	11	Ambiguous: few impacts identified; the most recurrent is a negative one	Q21–Q22
	Services delivery	15	Clear: agreement on the improvement in service quality; fear the increasing demand for screening	Q23–Q24

* Examples of quotes available in Appendix A.

## Data Availability

The data are not available for ethical reasons. The consent form explicitly stated that the results of the research would only be presented in aggregate form and individual participant results would never be released.

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
