# Peer review of "Toward a Population-Based Breast Cancer Risk Stratification Approach? The Needs and Concerns of Healthcare Providers"

_jpm, 2021, doi:10.3390/jpm11060540_

Round 1
Reviewer 1 Report
This study explores important issues around the implementation of risk-based screening. It is well written and findings will be of interest to readers. I have several minor suggestions:
- Reduce the number of acronyms (especially, HP, PQDCS, KM).
- Please add more information about the limitations related to the timing of data collection. The authors mention that it's been a few years; please me more precise: the data are now over 5 years old. Describe the ways in which qualitative data may have changed and provide citations/evidence for your hypotheses.
Author Response
Please see the attachement.

Reviewer 2 Report
Thank you for the opportunity to review. Below please find my comments.
- The interview only based on 15 interviews. It is not clear if saturation has been reached.
- Specific numbers of respondents should be included instead of just saying "most respondents" or "majority".
- Suggest authors to seek English editor's help. The wording and sentence structure made it hard to comprehend the text from time to time.
